# Sarcopenia Risk Evaluation in a Sample of Hospitalized Elderly Men and Women: Combined Use of the Mini Sarcopenia Risk Assessment (MSRA) and the SARC-F

**DOI:** 10.3390/nu13020635

**Published:** 2021-02-16

**Authors:** Andrea P. Rossi, Cesare Caliari, Silvia Urbani, Francesco Fantin, Piero Brandimarte, Angela Martini, Elena Zoico, Giulia Zoso, Alessio Babbanini, Alfredo Zanotelli, Mauro Zamboni

**Affiliations:** Department of Medicine, Division of Geriatric, University of Verona, 37126 Verona, Italy; caliaricesare@gmail.com (C.C.); silviaurbani93@gmail.com (S.U.); francesco.fantin@univr.it (F.F.); brandimarte.p@gmail.com (P.B.); martiniangela89@gmail.com (A.M.); elena.zoico@univr.it (E.Z.); zoso90@live.it (G.Z.); alessiobabba@gmail.com (A.B.); alfredo.zanotelli@outlook.com (A.Z.); mauro.zamboni@univr.it (M.Z.)

**Keywords:** sarcopenia, muscle strength, screening, physical limitations

## Abstract

Background: SARC-F and Mini Sarcopenia Risk Assessment (MSRA) questionnaires have been proposed as screening tools to identify patients at risk of sarcopenia. The aim of this study is to test the use of SARC-F and MSRA, alone and combined, as a pre-screening tool for sarcopenia in geriatric inpatients. Methods: 152 subjects, 94 men and 58 women, aged 70 to 94, underwent muscle mass evaluation by dual energy X-ray absorptiometry (DXA), muscle strength evaluation by handgrip, and completed the MSRA, SARC-F and Activity of daily living (ADL) questionnaires. Results: 66 subjects (43.4%) were classified as sarcopenic according to the European Working Group on Sarcopenia in Older People 2 (EWGSOP2) criteria. The 7-item SARC-F and MRSA and 5-item MSRA showed an area under the curve (AUC) of 0.666 (95% confidence interval (CI): 0.542–0.789), 0.730 (95% CI: 0.617–0.842) and 0.710 (95% CI: 0.593–0.827), respectively. The optimal cut-off points for sarcopenia detection were determined for each questionnaire using the Youden index method. The newly calculated cut-off points were ≤25 and ≤40 for MSRA 7- and 5-items, respectively. The ideal cut-off for the SARC-F was a score ≥3. Applying this new cut-off in our study population, sensitivity and specificity of the 7-item MSRA were 0.757 and 0.651, and 0.688 and 0.679 for the 5-item MSRA, respectively. Sensitivity and specificity of SARC-F were 0.524 and 0.765, respectively. The combined use of the 7-item SARC-F and MSRA improved the accuracy in sarcopenia diagnosis, with a specificity and sensitivity of 1.00 and 0.636. Conclusion: 7-item SARC-F and MSRA may be co-administered in hospital wards as an easy, feasible, first-line tool to identify sarcopenic subjects.

## 1. Introduction

Sarcopenia is a clinical phenomenon characterized by progressive and generalized loss of skeletal muscle mass and strength, with a risk of unfavorable health consequences such as worsening disability, reduction of functional autonomy, increased risk of hospitalization, low quality of life and increased mortality [1,2,3,4,5] Identifying older adults with sarcopenia in different clinical settings is crucial, in order to put a brake on progression towards disability and other adverse health outcomes. The revised criteria for confirmed sarcopenia diagnosis of the European Working Group on Sarcopenia (EWGSOP2) require the presence of low muscle mass associated with reduced muscle function [1]. Muscle mass quantification involves a wide range of techniques, such as dual energy X-ray absorptiometry (DXA), computer tomography and magnetic resonance, which, however, are difficult to access, especially in primary care settings [5]. Therefore, there is a need for easy, feasible, and broadly available screening tools for sarcopenia.

Recently, questionnaires have been proposed as screening tools to identify patients at risk of sarcopenia.

Advanced age, reduced protein calorie intake, reduced physical activity, weight loss and recurrent hospitalizations are currently acknowledged as risk factors for sarcopenia. The Mini Sarcopenia Risk Assessment (MSRA) questionnaire, evaluating risk factors for muscle mass and muscle strength loss, was proposed and validated as a screening tool for sarcopenia by Rossi et al. [6]. As a pioneer of sarcopenia screening tools, SARC-F, a test that measures parameters mostly related to self-reported disability, has greater consensus and broader diffusion [7].

In order to evaluate a screening test for sarcopenia, we have to weigh the implications of an error and therefore the costs of false positives and false negatives in the considered population.

We previously showed that for this condition, the relative cost of a false negative is higher than the cost of a false positive and therefore, we need to locate the threshold that maximizes the sum of sensitivity and specificity, favoring high sensitivity, rather than high specificity [6].

Yang et al. [8] tested the validity of the MSRA questionnaire, comparing it with SARC-F, on a population of 384 community-dwelling older adults.

The subjects were therefore classified into sarcopenic and non-sarcopenic according to the Asian Working Group for Sarcopenia (AWGS) criteria. In their study, SARC-F showed a sensitivity of 0.295 and specificity of 0.981, while MSRA, in its two forms, showed a sensitivity and specificity of 0.869 and 0.396 in the 7-item version, and sensitivity of 0.902 and specificity of 0.706 in the 5-item version. The diagnostic accuracy of the 3 tests was assessed using the area below the receiving operating characteristics ROC curve, which was 0.89 for SARC-F, 0.7 for 7-item MSRA and 0.85 for 5-item MSRA. According to the authors, therefore, the study showed that MSRA testing is a good screening tool for sarcopenia. In comparison with SARC-F, this questionnaire showed a similar diagnostic accuracy, with a better sensitivity.

To the best of our knowledge, no previous study evaluated the predictive value of the MRSA questionnaire in comparison with the SARC-F questionnaire for sarcopenia diagnosis in a sample of hospitalized elderly males and females.

The primary objective of the study is to estimate the sensitivity and specificity of the SARC-F and MSRA questionnaires in order to assess the ability to identify sarcopenia according to the EWGSOP2 criteria, in subjects admitted to a geriatric acute care ward.

The secondary objective is to test if the combined use of SARC-F and MSRA as a pre-screening tool can improve the accuracy in detecting sarcopenic subjects in this population.

## 2. Materials and Methods

### 2.1. Subjects

Subjects included in this study were recruited from patients admitted to the Department of Geriatrics at the University Hospital of Verona.

Subjects with age >65 years of both sexes, able to express informed consent, were included in the study.

The exclusion criteria included serious clinical conditions that precluded the sarcopenia screening test administration (handgrip-test or DXA), individuals with acute renal failure, and severe chronic renal failure or heart failure in the acute phase were also excluded.

In total, 94 men and 58 women aged 66 to 94 were recruited. All subjects underwent clinical evaluation, with a collection of structured anamnesis with particular attention to the reason for hospitalization and concomitant pathologies.

### 2.2. Anthropometry

Subjects underwent body weight measurement (Salus scale, Milan, Italy). The height for patients was measured using a stadiometer, with an approximation of 0.5 cm (Salus Stadiometer Milan, Italy). The body mass index (BMI) was calculated as the ratio between weight and height squared (kg/m^2^).

### 2.3. Body Composition

The appendicular fat-free mass was determined using DXA (Hologic Horizon W, software version 13.6.0.4). The physical characteristics and concepts of DXA measurement have been described elsewhere [9]. Daily quality checks were carried out following the manufacturer’s instructions. All scans were subsequently analyzed by a single trained operator.

### 2.4. Muscle Strength

Muscle strength was assessed by handgrip dynamometer (Jamar Handheld Dynamometer, Sammons Preston Rolyan, IL, USA) using a standardized protocol [10]. The best of three trials with the dominant hand was used for the present analysis [11]. The cut-off used to evaluate strength reduction was <27 kg for men and <16 kg for women.

### 2.5. Definition of Sarcopenic Subjects Based on EWGSOP2 Criteria

According to the EWGSOP2 criteria, sarcopenia was defined as the presence of low muscle mass plus low muscle strength [1]. Appendicular skeletal muscle mass was converted to skeletal muscle index (SMI) by dividing it by height expressed in meters squared (kg/m^2^), as previously suggested by Baumgartner [12]. The cut-off values used to identify subjects with low muscle mass were defined at <7.0 kg/m^2^ for men and <5.5 kg/m^2^ for women [1].

### 2.6. SARC-F and Mini Sarcopenia Risk Assessment Questionnaire

Subjects were administered questionnaires to assess the risk of sarcopenia.

The SARC-F questionnaire is composed of five domains: strength, assistance with walking, rising from a chair, climbing stairs and falls [7]. A score of ≥4 out of 10 points indicates a risk of sarcopenia.

The MRSA questionnaire, in its two forms, the 5- and 7-items, investigates 7 parameters regarding general and anamnestic assessment and nutritional assessment [6].

The 7-item MSRA consists of four questions related to general assessment (age, physical activity level, number of hospitalizations in the previous year and weight loss) and three related to dietary assessment (three questions related to meals number per day, milk and dairy products consumption and protein consumption). A score >30 was considered as normal, whilst a score ≤30 indicative of risk of sarcopenia, as previously described [6]. The 5-item MSRA does not include questions about the number of meals and consumption of milk and dairy products, and subjects scoring ≤45 are at risk of sarcopenia. More details are described elsewhere [6].

### 2.7. Covariates

Sociodemographic variables (age, gender, smoking habits, education) were assessed from a clinical interview at hospital admission. Functional status in basic activities of daily living (ADL) was assessed using the Katz ADL Index [13]. This questionnaire assesses basic activities of daily living such as transferring, continence, feeding, bathing, dressing and toileting, [13]. The subjects were assessed for independence in six functions of activities by receiving a score of Yes or No. Scores range from 6, indicating total independence, to 0, indicating complete dependence.

Weight loss and appetite decrease with consequent reduction of caloric intake (0 = no, 1 = yes, mild, 2 = yes, moderate to severe) were also recorded.

Diagnoses of specific medical conditions were gathered from the patient, attending physicians and through a careful review of medical charts.

Information regarding the comorbidities of the enrolled patients was retrospectively retrieved from the patient charts using manual reviews of the clinical history.

The Charlson Comorbidity Index was calculated using the appropriate weights. The definitions of the variables were those used in the Charlson Comorbidity Index [14]. The Charlson Age Comorbidity Index was also calculated by adding the age-adjusted comorbidity index. Moreover, the 30-point Folstein Mini Mental State Examination (MMSE) was administered in-person by a trained physician [15].

### 2.8. Statistical Analysis

Results are reported as means ± standard deviation (SD) and percentages. Collected data were summarized using descriptive statics.

Study population was divided into sarcopenic and non-sarcopenic subjects based on diagnostic criteria for confirmed sarcopenia proposed by the EWGSOP2. T-test for unpaired data was used to compare means in sarcopenic and non-sarcopenic subjects. Percentages were compared employing Fisher’s exact test.

The overall ability of the 7- and 5-item MSRA and SARC-F to discriminate subjects with sarcopenia was evaluated by means of the receiver-operating characteristic (ROC) curve, obtained by plotting estimates of the true positive rate (i.e., sensitivity) against the false positive rate (i.e., 1–specificity) for each possible score of the MSRA questionnaire. Specificity was defined as the proportion of non-cases (non-sarcopenic subjects) with SARC-F or MSRA score above the threshold considered, while sensitivity was defined as the proportion of cases (sarcopenic subjects) with SARC-F or MSRA score equal to or below the threshold considered. The area under the ROC curve (AUC) was calculated as a summary measure of the overall diagnostic accuracy across the spectrum of test values, as described in Fawcett [16]. Confidence intervals for the AUC, as well as comparisons between ROC curves, were performed according to DeLong et al. [17]. MSRA and SARC-F optimal threshold points were determined using the Youden index, defined as the point that maximizes the sum of sensitivity and specificity [18].

Using the cut-off values obtained, the sensitivity and specificity were calculated for each test and for the combination of both MSRA versions with SARC-F, and the statistical significance was calculated by means of a chi-square test.

A statistical significance level of 0.05 was considered in all statistical analyses. All statistical analyses were performed using SPSS (version 21.0 for Windows) [19].

## 3. Results

In total, 94 men and 58 women with mean age of 81.13 ± 5.53 years were involved in the study: 66 out of 152 subjects, 43.4% of the study population, were classified as sarcopenic according to the EWGSOP2 criteria.

Baseline characteristics of the study population (mean ± SD) are presented in Table 1. Sarcopenic subjects showed significantly higher age, weight and ADL score as compared to non-sarcopenic subjects. Again, scores in the 5- and 7-item MSRA, SARC-F as well as BMI were significantly lower in sarcopenic subjects.

In order to compare the predictive power, the ROC curve and the area under the curve of each questionnaire were calculated (Figure 1). The SARC-F questionnaire showed an area under the curve (AUC) of 0.666 (95% confidence interval (CI): 0.542–0.789). MSRA ROC curves showed significantly higher AUCs compared to SARC-F, respectively 0.730 (95% CI: 0.617–0.842) for the 7-item MSRA and 0.710 (95% CI: 0.593–0.827) for the 5-item MSRA.

To evaluate the optimal cut-off values in our study population, the Youden index was calculated for the scores of the 7- and 5-item MSRA questionnaires and for the SARC-F questionnaire (Table 2).

Using the Youden index method, the optimal cut-off points to identify sarcopenia in our study population corresponded to a total score ≤25 and ≤40 for MSRA 7- and 5-items, respectively. The ideal cut-off for the SARC-F was a score ≥3.

Therefore, the sensitivity, specificity, positive and negative predictive value of the tests, considering the different MSRA and SARC-F cut-offs obtained using the Youden index in the study population, were recalculated (Table 3). The 7- and 5-item MSRA questionnaires showed sensitivity and specificity of 0.757 and 0.651 for the 7-items and 0.688 and 0.679 for the 5-items, respectively. The SARC-F questionnaire showed specificity of 0.765 and sensitivity of 0.524. Combining the SARC-F and 7-item MSRA, specificity and sensitivity were respectively 1.00 and 0.636. By combining SARC-F and the 5-item MSRA, specificity and sensitivity were respectively 0.977 and 0.424.

## 4. Discussion

Our study shows that MRSA is a useful tool for sarcopenia risk assessment, showing high accuracy in identifying sarcopenic subjects in a population of hospitalized elderly adults. Furthermore, MSRA, as combined with SARC-F, reduced the number of sarcopenic subjects with negative tests.

The main strength of this study is that we compared the accuracy of SARC-F and MSRA questionnaires in identification of sarcopenic subjects using DXA, the gold standard for muscle mass evaluation, applying the most recent European criteria for sarcopenia.

In our study population, both questionnaires, SARC-F and MRSA, showed high sensitivity and specificity, but only MSRA had an AUC value above 0.7, which suggests a moderate level of diagnostic accuracy. SARC-F is currently validated in many different languages and settings. It differs from the MSRA questionnaire, as it is based on a subjective and non-measurable assessment of disability, referred by the patient or by the caregiver. MSRA, on the other hand, is based on objective and measurable parameters selected on the basis of review of the literature regarding risk factors for sarcopenia: age, hospitalization, eating habits, protein intake, level of physical activity and weight loss.

In the SARC-F questionnaire, the items chosen are associated with the subject’s general disability, and this tool showed good association with mortality and incident disability [20].

In our study, it emerged that both forms of the MSRA, the 7-item and the 5-item, show higher accuracy, and in particular, higher sensitivity, compared to SARC-F, in identifying sarcopenic subjects in a population of hospitalized elderly men and women. This is particularly important since for sarcopenia, the relative cost of a false negative is higher than the cost of a false positive [6] and then sensitivity should be privileged for a reliable and valid sarcopenia screening test. Our data seems to indicate that the MSRA questionnaire shows high sensitivity, as compared with the SARC-F, making it suitable as a first-line screening tool for sarcopenia also in hospitalized subjects.

This is partially in contrast with the results obtained by Yang et al. [21], who demonstrated in community-dwelling elderly, randomly recruited from the general population, that SARC-F exhibits a diagnostic accuracy similar to MSRA. It should be noted that in their study, however, muscle mass evaluation was obtained by bioimpedentiometry (BIA), whilst in our study, it was measured with DXA, a more accurate technique. In addition, the two populations significantly differ in terms of age, ADL score, race and sarcopenia diagnosis applied criteria. In fact, Yang applied the Asian Working Group criteria for sarcopenia.

Applying the Youden index to our study population, we showed that the ideal SARC-F cut-off for sarcopenia diagnosis is lower than the score proposed in the validation study [7]. However, this result is in line with a previous report from Li et al. [22] and was expected, considering that our study population, recruited in a geriatric acute care ward, is older and has larger burden of diseases.

The MSRA cut-offs previously proposed by our group for the diagnosis of sarcopenia for the general population [6] appear to maintain high sensitivity values also in hospitalized subjects, albeit burdened by a limited reduction in specificity. Therefore, MSRA proves to be a good first-line screening tool to identify the presence of sarcopenia also in a population of elderly hospitalized subjects, and the use of lower thresholds calculated with the Youden index only slightly improves specificity, with non-significant differences in sensitivity.

The combined use of the 7-item MSRA and SARC-F increased the specificity in detecting sarcopenia, with a concomitant limited reduction of sensitivity.

To the best of our knowledge, only few studies have evaluated the reliability of sarcopenia questionnaires in an acute care ward and no previous studies have tested the predictive value of the combination of both questionnaires [23,24,25]. The best combination of sensitivity and specificity values for sarcopenia diagnosis in our population of hospitalized subjects was obtained at 22.5 points, corresponding to a score lower than 25 in the 7-item MSRA, and at 37.5 points, corresponding to a score lower than 40 in the 5-item MSRA. This is in line with the results of Kryinska-Siemaszko et al., who showed that the MSRA questionnaire, also exploring nutritional deficiencies and their role in sarcopenia development, requires a lower cut-off for both the 7- and 5-items when applied to a population of subjects with high prevalence of malnutrition [24].

In a very recent study involving elderly individuals with cardiac heart failure, Zhao et al. found that SARC-F and MSRA showed similar accuracy in finding subjects with sarcopenia [25].

The lack of accuracy shown in the Asian population can be explained by the fact that most Chinese individuals do not eat dairy products every day, and some do not even eat dairy products at all.

Our data seem to suggest that the combined use of SARC-F and MSRA reduces the number of non-sarcopenic subjects with positive tests, with a small increase in the number of sarcopenic subjects with negative tests, and with higher sensitivity when SARC-F was combined with the 7-item MSRA than with the 5-item MSRA instead. This result should be confirmed in further studies with larger samples and in different clinical settings.

In our study, the prevalence of sarcopenia was found to be particularly high. This finding is explained by the characteristics of the study population, which was composed of elderly, hospitalized and comorbid subjects, and by the method used for muscle mass evaluation.

This high prevalence is only partially in line with other studies that have assessed the presence of sarcopenia in hospitalized patients. Our study group, in a similar population of 119 patients with an average age slightly lower (80.4 years), hospitalized in a geriatric ward [26], found an incidence of sarcopenia in about 1 patient out of 4. However, the muscle mass was assessed by bioimpedentiometry (BIA), which is less sensitive than DXA used in our study to identify sarcopenic subjects. In another study with BIA, Bianchi et al. [27], in a population of 655 hospitalized patients, found a prevalence of sarcopenia close to 35%. However, their study population also included non-geriatric patients, and the authors themselves found a higher prevalence of sarcopenia among older subjects, similar to our study population.

Several studies previously showed that the agreement between DXA and BIA is acceptable, although it is known that BIA tends to overestimate appendicular muscle mass compared to DXA [28,29], with measured difference from 1 to 3 kg. Moreover BIA, also in comparison with the “gold standard” computed tomography [3], tends to overestimate skeletal muscle mass because it cannot discriminate among appendicular, non-appendicular fat and non-fat mass.

Some limitations of this study warrant a mention.

Firstly, the sample size (152 subjects) is reduced and may not be representative of the whole population of hospitalized elderly subjects, considering that subjects unable to independently lay down on the DXA table were excluded.

Secondly, the encouraging combined use of both questionnaires in the same population is limited to hospitalized subjects and should also be confirmed in community-dwelling elderly people and long-term care facilities.

Thirdly, in this analysis, we did not take into account the severe sarcopenia definition, also characterized by low physical performance [1]. However, a recent study showed that only 1.8% of hospitalized subjects of similar age have been traced back under this definition [30].

Lastly, although only few subjects had mild cognitive impairment in our study population, the potential impact of dementia on self-administrated questionnaires, such as MMSRA and SARC-F, must be recognized. Further studies involving caregivers would be needed to test the validity of sarcopenia screening tools in this population.

## 5. Conclusions

The data of this study show that both the MRSA and the SARC-F questionnaires can be used as a valid first-level screening tool for the risk of sarcopenia, even in a population of hospitalized patients. The use of questionnaires as a pre-screening tool for sarcopenia has been previously strongly promoted in the revised European guidelines for sarcopenia. Our data seems to suggest the combined use of both questionnaires, SARC-F and the 7-item MSRA, in order to reduce the number of false positives, at least for inpatients. SARC-F and MSRA questionnaires could be co-administered easily, quickly and economically in hospital wards, as a first-line tool to identify sarcopenic subjects at greater risk of exceeding the disability threshold during their hospital stay.

## Figures and Tables

**Figure 1 nutrients-13-00635-f001:**
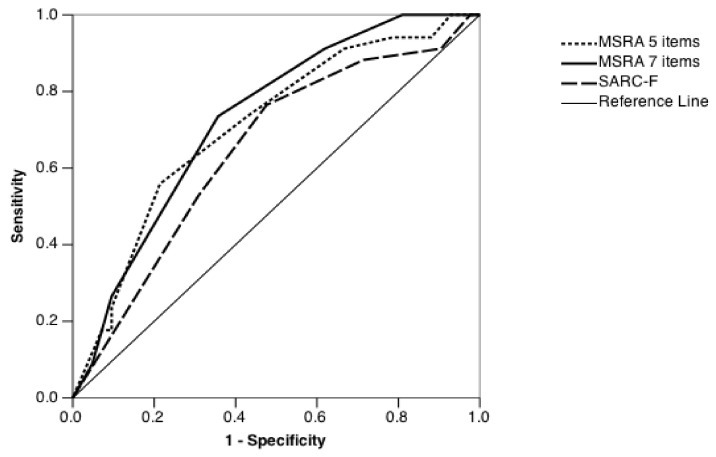
ROC curve associated with SARC-F, 7-item MSRA and 5-item MSRA score. MSRA = Mini Sarcopenia Risk Assessment. ROC = receiving operating characteristics; MSRA = Mini Sarcopenia Risk Assessment.

**Table 1 nutrients-13-00635-t001:** Characteristics of the study population (*n* = 152) in sarcopenic and non-sarcopenic subjects according with the EWGSOP2 definition.

	Non Sarcopenic (*n* = 86)		Sarcopenic (*n* = 66)		
	Mean ± SD	Min–Max	Mean ± SD	Min–Max	*p*
Age (years)	79.79 ± 5.47	66–93	82.88 ± 5.18	70–94	= 0.015
Gender (male)	60 (69.8%)		34 (51.5%)		N.S.
Weight (kg)	76.36 ± 14.03	51–111	64.39 ± 12.06	39–91	<0.001
Height (m)	1.66 ± 0.08	1.52–1.80	1.62 ± 0.08	1.49–1.80	= 0.022
BMI (kg/m^2^)	27.40 ± 4.15	26.95–29.85	24.33 ± 3.74	23.15–25.21	= 0.001
ADL score	5.4 ± 0.4	1–6	5.0 ± 0.3	1–6	<0.05
MMSE	26.55 ± 3.3	21.70–30	26.69 ± 2.80	21.30–30	N.S.
Charlson Comorbidity Index	2.06 ± 1.56	0–8	2.75 ± 2.07	0–9	<0.05
Charlson Age Comorbidity Index	5.80 ± 1.83	2–12	6.53 ± 2.32	2–13	<0.05
7-item MSRA score	23.25 ± 7.06	0–35	18.93 ± 6.58	10–35	<0.01
5-item MSRA score	35.81 ± 12.43	0–55	27.87 ± 13.17	0–55	<0.01
SARC-F score	1.60 ± 1.52	0–6	3.39 ± 2.22	0–6	<0.05
7-item MSRA score ≤25 (%)	16 (18.6%)		50 (75.7%)		<0.001
5-item MSRA score ≤40 (%)	44 (51.2%)		52 (78.8%)		<0.05
SARC-F score ≥3 (%)	18 (20.1%)		42 (63.6%)		<0.001
Combined SARC-F/7-item MSRA (%)	0 (0.0%)		42 (63.6%)		<0.001
Combined SARC-F/5-item MSRA (%)	2 (2.3%)		28 (42.4%)		<0.001
Handgrip (kg)	29.40 ± 8.16	10.50−42.50	18.88 ± 5.50	6.50−26.80	<0.001
Gait-speed (<0.8 m/s)	14 (41.2%)		39 (92.9%)		<0.001
SMI (kg/m^2^)	6.85 ± 1.09	6.86–7.52	5.63 ± 0.74	4.23–5.83	<0.001

BMI = body mass index; MSRA = Mini Sarcopenia Risk Assessment; EWGSOP2 = European Working Group on Sarcopenia in Older People 2; SMI = skeletal muscle index; ADL = activity of daily living.

**Table 2 nutrients-13-00635-t002:** Cut-off for sarcopenia of different questionnaires calculated with the Youden Index.

7-item MSRA			
Score	Sensitivity	1–Specificity	Youden Index
2.5	0.024	0	0.024
7.5	0.048	0	0.048
12.5	0.190	0	0.190
17.5	0.381	0.088	0.293
22.5	0.643	0.265	0.378
27.5	0.905	0.735	0.170
32.5	0.952	0.912	0.040
36	1	1	0
5-item MSRA			
Score	Sensitivity	1–Specificity	Youden Index
5	0.071	0	0.071
12.5	0.119	0.059	0.06
17.5	0.214	0.059	0.155
22.5	0.333	0.088	0.245
27.5	0.571	0.265	0.306
32.5	0.714	0.382	0.332
37.5	0.786	0.441	0.345
42.5	0.905	0.765	0.140
47.5	0.905	0.824	0.081
52.5	0.929	0.824	0.105
56	1	1	0
SARC-F			
Score	Sensitivity	1–Specificity	Youden Index
0.5	0.881	0.706	0.175
1.5	0.69	0.471	0.219
2.5	0.524	0.235	0.289
3.5	0.333	0.118	0.215
4.5	0.286	0.118	0.168
5.5	0.095	0.088	0.007
6.5	0.048	0.029	0.019
7.5	0.024	0	0.024

MSRA = Mini Sarcopenia Risk Assessment; SARC-F.

**Table 3 nutrients-13-00635-t003:** Cut-off for sarcopenia of different questionnaires calculated with the Youden Index.

	Specificity	Sensitivity	Negative Predictive Value	Positive Predictive Value
7-item MSRA score ≤30	0.186	0.879	0.671	0.454
7-item MSRA score ≤25 (new cut-off)	0.651	0.757	0.777	0.624
5-item MSRA score ≤45	0.163	0.909	0.700	0.454
5-item MSRA score ≤40 (new cut-off)	0.679	0.688	0.739	0.622
SARC-F score ≥4	0.651	0.576	0.667	0.559
SARC-F score ≥3 (new cut-off)	0.765	0.524	0.677	0.631
Combined SARC-F/7-item MSRA	1.00	0.636	0.782	1.00
Combined SARC-F/5-item MSRA	0.977	0.424	0.689	0.934

## Data Availability

Data are not available since this option was not included in the originally signed consent.

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
