# Peer review of "Sarcopenia Risk Evaluation in a Sample of Hospitalized Elderly Men and Women: Combined Use of the Mini Sarcopenia Risk Assessment (MSRA) and the SARC-F"

_nutrients, 2021, doi:10.3390/nu13020635_

Round 1
Reviewer 1 Report
I would like to thank the reviewers for allowing me to review this manuscript. The manuscript is well-written, but I have the following concerns
- This study included elderly patients admitted to the geriatrics department of a university hospital, but the disease was not controlled. Since sarcopenia is greatly influenced by disease, the validity of the results of the sarcopenia risk assessment in this patient group is questionable. The authors should control the disease adequately. The authors should identify the patient's primary disease and comorbidities and control for these in the analysis.
- There is not enough description of dementia. There are doubts about the validity of the results of the questionnaire for patients with dementia.
- Please explain in detail how the authors assessed the ADL score.
Author Response
Reviewer 1 Comments and Suggestions for Authors
I would like to thank the reviewers for allowing me to review this manuscript. The manuscript is well-written, but I have the following concerns.
- This study included elderly patients admitted to the geriatrics department of a university hospital, but the disease was not controlled. Since sarcopenia is greatly influenced by disease, the validity of the results of the sarcopenia risk assessment in this patient group is questionable. The authors should control the disease adequately. The authors should identify the patient's primary disease and comorbidities and control for these in the analysis.
We thank the Reviewer for his comment and we completely agree that sarcopenia is influenced by comorbidity. The main aim of the study was to estimate the ability of SARC-F and MSRA questionnaires to identify sarcopenia in hospitalized inpatient according with EWGSOP2 criteria. However, main comorbidity information are available for our study population and we calculated the Charlson Index and the Charlson Age Index. In accordance with the Reviewer suggestion we added this information in Table 1. In particular, as expected, we found that the sarcopenic group had higher comorbidity. On the contrary we didn’t found differences in Charlson Index and Charlson Age Index using SARC-F and MSRA threshold to classify our study population.
- There is not enough description of dementia. There are doubts about the validity of the results of the questionnaire for patients with dementia.
We completely agree with the Reviewer comment. Fortunately, the Mini Mental State Examination was performed in our population and we added this measurement in the text and in Table 1. The MMSE score was not different in subjects with or without Sarcopenia. Furthermore, we have to consider that our population included only subjects able to express informed consent and that subjects with serious clinical conditions that precluded the sarcopenia screening test administration (handgrip-test or DXA) were excluded. As a result, only few subjects had a MMSE lower than 24. However, we recognize the potential impact of dementia on self administrated questionnaires and accordingly we added a comment in the study limitations.
Page 5
“Moreover the 30-point Folstein MMSE was administered in-person by a trained physician (15).”
Page 13
“Lastly, although only few subjects had mild cognitive impairment in our study population, the potential impact of dementia on self administrated questionnaires, such as MMSRA and SARC-F, must be recognized. Further studies involving caregivers would be needed to test the validity of sarcopenia screening tools in this population.”
- Please explain in detail how the authors assessed the ADL score.
In agreement with the Reviewer comment we added the details on how the ADL score was assessed in our population in the Methods section of the revised version of the manuscript.
Page 4
“Functional status in basic activities of daily living (ADL) was assessed using the Katz ADL Index (13). This questionnaire assesses basic activities of daily living such as transferring, continence, feeding, bathing, dressing, and toileting, (13). The subjects has been assessed for independence in six functions of activities by receiving a score of Yes or No. Scores range from 6, indicating total independence to 0, indicating complete dependence.”
Reviewer 2 Report
This is an interesting study but the authors could follow these suggestions:
- Please discuss in the introduction and the discussion the importance of favouring specificity or sensitivity in sarcopenia screening.
- Based on the previous comment, the Younden index method is probably not the best to find the optimal cut-off value.
- In the abstract please provide the new Se and Sp for the SARC-F new cut-off.
- Please provide a full table with the Sp, the Se, the negative predictive value and positive predictive value for all screening tools with the old and the new cut-offs as well as the various combinations.
- Please clearly state in the text what are the new cut-off values.
Author Response
This is an interesting study but the authors could follow these suggestions:
- Please discuss in the introduction and the discussion the importance of favouring specificity or sensitivity in sarcopenia screening.
We thank the Reviewer for this comment.
In order to understand if we have to privilege specificity or sensitivity, we have to evaluate the implications of an error, and therefore the costs of false positives and false negatives in the elderly population considered.
The cost of a false positive, considering both, instrumental and clinical evaluation, is 143.08 euro in Italy. Otherwise the false negative costs, based on the one-month health related costs of a sarcopenic subject compared to non sarcopenic subject in a similar Caucasian European population, the Dutch population of the MaSS study, equal to 930 euro (Mijnarends DM, Schols JMGA, Halfens RJG, Meijers JMM, Luiking,YC, Verlaane S, Evers SMAA. Burden-of-illness of Dutch community-dwelling older adults with sarcopenia: Health related outcomes and costs. Eur J Geriatr 2016; 7: 276–284).
Therefore we need to locate the threshold that maximize the sum of sensitivity and specificity and in the same time guarantee high sensitivity.
We added accordingly comments in the introduction and in the discussion of the revised version of the manuscript.
Page 2
“In order to evaluate a screening test for sarcopenia, we have to weight the implications of an error and therefore the costs of false positives and false negatives in the considered population.
We previously showed that for this condition the relative cost of a false negative are higher than the cost of a false positive and therefore we need to locate the threshold that maximize the sum of sensitivity and specificity, favoring high sensitivity, rather than high specificity (6).”
Page 11
“In our study emerged that both forms of the MSRA, the 7-items and 5-items, show higher accuracy and in particular, higher sensitivity, compared to SARC-F, in identifying sarcopenic subjects in a population of hospitalized elderly men and women. This is particularly important since for sarcopenia the relative cost of a false negative are much higher than the cost of a false positive (6) and then sensitivity should be privileged for a reliable and valid sarcopenia screening test. Our data seems to indicate that the MSRA questionnaire shows high sensitivity, as compared with the SARC-F, making it suitable as a first line screening tool for sarcopenia also in hospitalized subjects.”
- Based on the previous comment, the Younden index method is probably not the best to find the optimal cut-off value.
See previous comment.
- In the abstract please provide the new Se and Sp for the SARC-F new cut-off.
In agreement with the Reviewer comment the manuscript has been modified accordingly.
Page 1
“The Sensitivity and specificity of SARC-F, were instead 0.524 and 0.765, respectively.”
- Please provide a full table with the Sp, the Se, the negative predictive value and positive predictive value for all screening tools with the old and the new cut-offs as well as the various combinations.
This information has been added in Table 3 in the Revised version of the manuscript.
- Please clearly state in the text what are the new cut-off values.
This information has been clearly stated in the Abstract and Results section of the revised version of the manuscript.
Page 1
“The newly calculated cutoff were ≤ 25 and ≤ 40 for MSRA 7- and 5-items, respectively. The ideal cut-off for the SARC-F was a score ≥ 3.”
Page 6
“Using the Youden index method, the optimal cut-off points to identify sarcopenia in our study population corresponded to a total score ≤ 25 and ≤ 40 for MSRA 7- and 5-items, respectively. The ideal cut-off for the SARC-F was a score ≥ 3.”
Round 2
Reviewer 1 Report
This manuscript has been appropriately revised in response to my previous comments.
Reviewer 2 Report
OK